# A simple mortality risk prediction score for viper envenoming in India (VENOMS): A model development and validation study

**Maya Gopalakrishnan**[1]*, **Suman Saurabh**[2], **Pramod Sagar**[3], **Chanaveerappa Bammigatti**[4], **Tarun Kumar Dutta**[5]

1 Department of Internal Medicine, All India Institute of Medical Sciences Jodhpur, Rajasthan, India,
2 Department of Community Medicine and Family Medicine, All India Institute of Medical Sciences, Jodhpur, Rajasthan, India, 3 Department of Cardiology, Madras Medical Mission, Chennai, Tamil Nadu, India,
4 Department of Medicine, Jawaharlal Institute of Medical Education and Research, Puducherry, India,
5 Department of Medicine, Mahatma Gandhi Medical College and Research Institute, Puducherry, India

* maya.gopalakrishnan@gmail.com

## Abstract

### Background

Snakebite is a neglected problem with a high mortality in India. There are no simple clinical prognostic tools which can predict mortality in viper envenomings. We aimed to develop and validate a mortality-risk prediction score for patients of viper envenoming from Southern India.

### Methods

We used clinical predictors from a prospective cohort of 248 patients with syndromic diagnosis of viper envenoming and had a positive 20-minute whole blood clotting test (WBCT 20) from a tertiary-care hospital in Puducherry, India. We applied multivariable logistic regression with backward elimination approach. External validation of this score was done among 140 patients from the same centre and its performance was assessed with concordance statistic and calibration plots.

### Findings

The final model termed VENOMS from the term "Viper ENvenOming Mortality Score included 7 admission clinical parameters (recorded in the first 48 hours after bite): presence of overt bleeding manifestations, presence of capillary leak syndrome, haemoglobin <10 g/dL, bite to antivenom administration time > 6.5 h, systolic blood pressure < 100 mm Hg, urine output <20 mL/h in 24 h and female gender. The lowest possible VENOMS score of 0 predicted an in-hospital mortality risk of 0.06% while highest score of 12 predicted a mortality of 99.1%. The model had a concordance statistic of 0·86 (95% CI 0·79–0·94) in the validation cohort. Calibration plots indicated good agreement of predicted and observed outcomes.

**Data Availability Statement:** All relevant data are within the manuscript and its Supporting Information files.

**Funding:** The authors received no funding for this work.

**Competing interests:** The authors have declared that no competing interests exist.

## Conclusions

The VENOMS score is a good predictor of the mortality in viper envenoming in southern India where Russell's viper envenoming burden is high. The score may have potential applications in triaging patients and guiding management after further validation.

## Author summary

More than 58,000 people die of snakebites each year in India. Russell's viper, saw scaled viper and pit vipers are widely distributed and medically important venomous snakes in India responsible for significant deaths and disabilities. Clinicians need easy-to-use bedside tools to make decisions on which patients are at a higher risk of dying after viper bites. In this study, conducted in Southern India, where Russell's viper is the commonest viper causing bites, we have evolved and validated a simple risk prediction score. This uses seven clinical and laboratory features to estimate a patient's risk of dying in the hospital due to the bite. The study showed that the score has good predictive ability when tested in a similar population of patients. We expect that the score is the first step in developing a tool that is likely to help health workers and doctors assess a patient's risk in primary-care peripheral or rural settings to help decide on early referral of high-risk patients who are likely to worsen.

## Introduction

Snakebite envenoming is a serious but neglected problem in the tropics [1,2]. South Asia, particularly, India has the largest burden of snakebite deaths and disabilities in the world [3,4]. Recent estimates suggest that annual mortality from snakebite envenomings in India is approximately 58,000; which is more than half the estimated global snakebite mortality. Thrice as many endure lifelong disabilities due to long-term consequences [4,5]. Affected are usually young adults belonging to lower socio-economic background who experience subsequent social stigma and discrimination [4,6,7]. In South Asia, the snake species under the epithet of "Big 4" i.e. *Daboia russelii*, *Echis carinatus*, *Bungarus caeruleus and Naja naja* garner widespread attention while the other regionally important snake species are also emerging as medically important [8,9]. The currently available polyvalent antivenom in India neutralizes venom of only these four species [10].

In clinical settings, snakebite envenoming syndromes are broadly categorized as neurotoxic and haemo-vasculotoxic. Neurotoxic symptoms are usually due to elapid bites i.e., cobra and krait, and vasculotoxic envenomings due to vipers. An important bedside test in establishing the diagnosis of viper envenoming is the whole blood clotting test (WBCT20) [11]. Two ml of freshly sampled venous blood in a dry, glass vessel or tube and left undisturbed for 20 minutes at ambient temperature. The vessel is tipped once, if the blood is still liquid (unclotted) and runs out, the patient is inferred to have hypofibrinogenaemia ("incoagulable blood") as a result of venom-induced consumption coagulopathy (VICC) [11].

Among the viper species, Russell's viper is widely distributed throughout Indian subcontinent including Sri Lanka and Myanmar [12,13]. A nationwide study and evidence synthesis estimated that 43% of reported bites in India are likely to be due to Russell's viper envenoming [4]. Russell's viper is reported as the species responsible for up to 80% mortality in several hospital-based series across India [14,15]. Variation in venom composition between Russell's

viper species from various parts of India leading to marked difference in neutralizing capability of the polyvalent antivenom has been recently demonstrated [12].

Russell's viper envenoming is clinically complex and challenging as it results in a rapidly progressive multisystem dysfunction culminating in mortality. The envenoming haemo-vasculotoxic syndrome affects platelets, coagulation factors (like factors V and X), endothelium of the vessel wall resulting in VICC, thrombotic microangiopathy and capillary leak syndrome (CLS) [16,17]. VICC presents with bleeding manifestations which can range from mild bleeding like gum bleeds and bite-site bleeding to life threatening bleeds such as intracranial haemorrhage and gastrointestinal bleeds [18]. CLS has been reported from Russell's viper bites in Southern India, Sri Lanka and Myanmar and is associated with a poor outcome [10,19]. CLS presents with manifestations of parotid swelling, conjunctival-chemosis, periorbital edema, hypotension, albuminuria and hemo-concentration. Other organ-systems including kidneys, heart, presynaptic neuromuscular junction and hypothalamo-pituitary axis are also affected in Russell's viper envenoming resulting in acute kidney injury (AKI), early neuromuscular paralysis, acute adrenal insufficiency and long-term consequences like chronic kidney disease and Sheehan like syndrome [20–24].

The other important viper species with widespread distribution is *Echis carinatus* which has two subspecies: *Echis carinatus* accounts for for envenomings in the Indian peninsula while *Echis carinatus sochureki* is thought to be responsible for bites in Northern India and Pakistan [25,26]. Echis envenoming presents with local swelling, coagulopathy and bleeding manifestations. Apart from this several pit vipers such as hump-nosed pit viper (*Hypnale hypnale)*, Himalayan and bamboo pit vipers in north-eastern India and Malabar pit viper in the western coast are also clinically significant [10]. Hump nosed pit viper can cause local necrosis, coagulopathy, bleeding and acute kidney injury and maybe misidentified as saw-scaled viper [27,28]. Syndromic diagnosis is widely applied especially in primary care settings despite its limitations in the absence of reliable species identification methods in routine clinical practice [29,30].

Managing viper bites is complicated involving multiple decisions like need for renal replacement therapy, ventilatory support for pulmonary edema, ionotropic support for distributive shock in capillary leak syndrome and transfusion support based on which organ systems are involved and when [31]. This is supported by several studies which report higher mortality and morbidity in viper envenoming [4,32]. Thus, viper envenomings have a complex pathogenesis with distinct prognostic factors involved implying that they merit the need for a distinct clinical decision support tool from elapid envenoming.

Recently, World health organization (WHO) has evolved a strategy to halve the snakebite mortality by 2030 as compared to 2015. One of the strategies in this call to action includes development of clinical decision support tools for improving outcomes [33]. Though several clinical parameters have been explored as mortality risk predictors in hospital-based studies, no simple score exists to quantify the prognostic factors affecting outcomes [34,35].

We aimed to develop and externally validate a simple, point-of-care mortality risk prediction score for patients presenting with syndromically diagnosed hemotoxic viper envenoming patients which could be potentially applied across healthcare settings.

## Methods

### Ethics statement

Both studies were approved by institutional ethics committees (JIPMER Institue Ethics Committee, JIP/IEC/SC/3/2012/13 and JIP/IEC/2014/1/24). Written informed consent was obtained at the time of data collection from the participant or parent/guardian along with

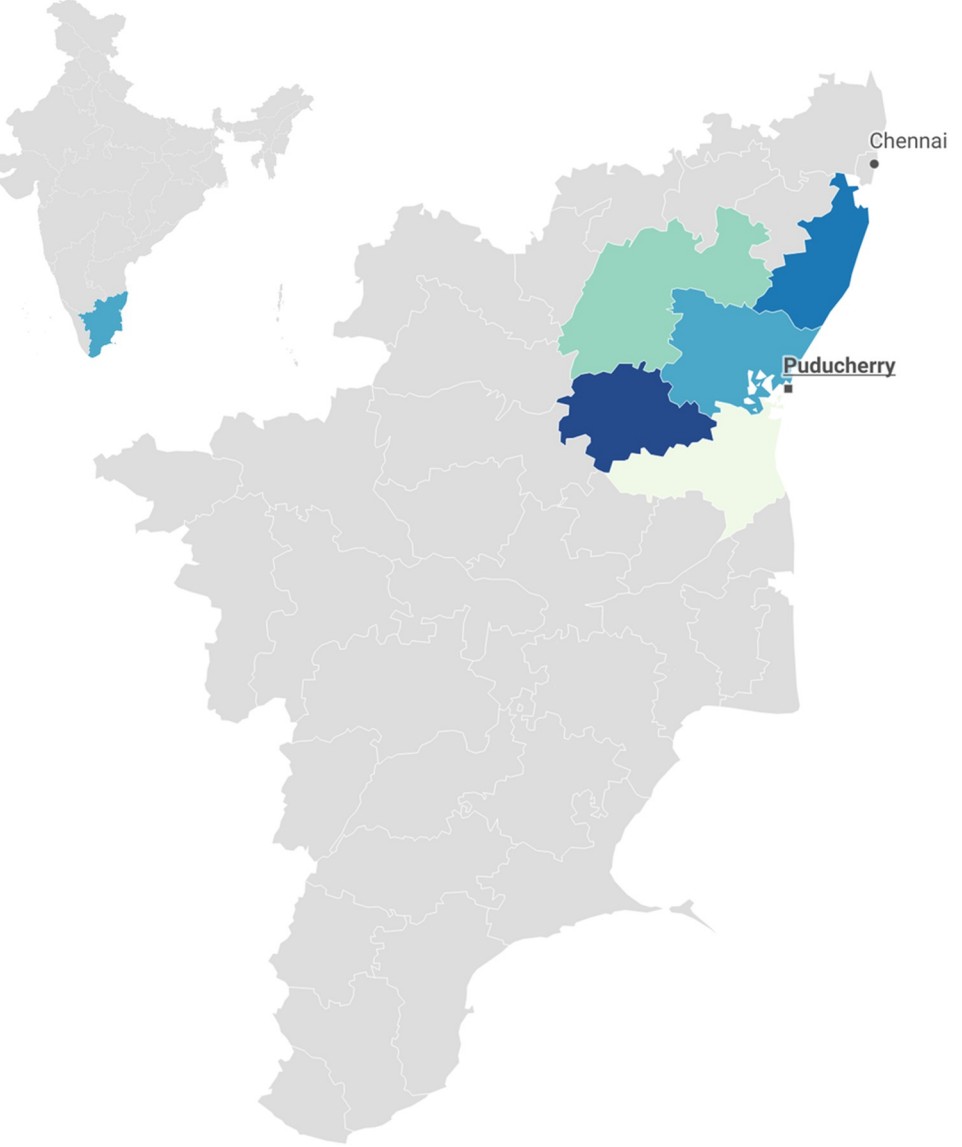

Map data: © OSM • Created with Datawrapper

**Fig 1. Map of India with state of Tamil Nadu.** The union territory of Puducherry (town), showing location of the study site with highlighted adjacent districts of the state of Tamil Nadu from where patients were enrolled. (Map not to scale. Maps created using https://www.datawrapper.de/).

participant's assent if the participant's age <18 years. However, repeat consent was not obtained, as this was a retrospective study using de-identified patient data from previous studies.

*Study setting, populations, and design cohorts*: The study site for both derivation and validation cohorts was a tertiary care referral hospital situated in Puducherry, India, located in the eastern coast of India. Our hospital has a catchment area of approximately 17,000 km$^2$ wherein 8 medically important snakes including the "big 4" have been routinely reported [36]. (Fig 1). Russell's viper is common and saw-scaled viper is routinely reported while pit vipers have not been reported in the area.

### Derivation cohort

We developed the model using data from a prospective cohort study of consecutive patients presenting to the emergency department of a tertiary care referral centre in Puducherry, India between September 2011 to August 2013.The clinical characteristics and outcomes of this prospective derivation cohort (n = 248) have been published previously [37]. Those patients ≥ 12 years of age, presenting with a history of snakebite or unknown bite with positive whole blood clotting test (WBCT20) and diagnosis of viper envenoming made by syndromic approach or identification of dead snake/photograph of the snake if brought by the patient were included. Syndromic diagnosis of viper envenoming was made based on syndromes 1, 2 and 5 in World Health Organization (WHO) guidelines. Syndrome 1 (All viperidae): Local envenoming (swelling) with bleeding/clotting disturbances. Syndrome 2: (Russell's viper in South India/ Myanmar/Sri Lanka): Local envenoming and bleeding/clotting disturbances with shock, acute kidney injury, conjunctival chemosis, acute pituitary insufficiency, ptosis, external ophthalmoplegia, facial paralysis or dark brown urine. Syndrome 5 (Russell's viper in Sri Lanka or South India): Bitten on land and paralysis with dark brown urine/acute kidney injury with bleeding/ clotting disturbances. Those with isolated neurotoxicity and local manifestations alone with normal WBCT20 were excluded (Syndromes 3 and 4) [31]. All patients in this cohort presented within 48 hours of bite while 67% presented within 6 hours of bite.

### Validation cohort

We validated the model in an external cohort of 140 patients who presented to the same centre from September 2013 to July 2015.This cohort was comprised of patients from a randomized clinical trial investigating two different doses of polyvalent antivenom [38]. This cohort included patients who had abnormal WBCT20 and syndromic diagnosis of viper envenoming. However, this cohort excluded those who had received greater than 200 mL (20 vials) antivenom prior to presentation (trial registered at CTRI/2015/05/005826). All patients in this cohort also presented within 48 hours of bite.

### Predictor variable selection

We searched for predictors of mortality in haemotoxic viper bite envenoming that were reported in previous studies or reviews (Table A in S1 Appendix -). We selected parameters that could easily be ascertained in different clinical settings with minimal interobserver variability and were part of the routine assessment in snakebite envenoming especially in primary care settings. Coagulation tests such as prothrombin time (PT), activated partial thromboplastin time (aPTT), serum fibrinogen, D-dimer were deliberately omitted considering the poor availability of these tests as point-of-care in primary care rural settings in India. For the purpose of this study clinical parameters assessed at 24 hours of admission, were defined as follows: a) signs of capillary leak syndrome (CLS) was defined as the presence of clinical evidence of at least one of the following: conjunctival chemosis, parotid swelling or periorbital puffiness with clinical evidence of pleural effusion or ascites b) overt bleeding: presence of bleeding from oral cavity, persistent bleeding from bite site hematuria, epistaxis, bleeding from intravenous puncture sites, hematemesis or melena, fresh bleeding per rectum, abnormal uterine bleeding or intracranial hemorrhage. c) renal dysfunction: Arbitrarily defined as serum creatinine > 3.0 mg/dl.) d) severe local envenoming: swelling involving more than one half of the bitten limb and bites involving the face/trunk. Urine output was measured over first 24 hours of admission and later converted to ml/hour.

Receiver operator characteristics (ROC) curves were constructed for each of the continuous variables from the derivation cohort to determine appropriate cut-offs to categorize them into

clinically significant categories (Table B in S1 Appendix). Categorization of continuous variables was done in order to simplify the final score. For identifying additional predictors, we performed univariable (unadjusted) logistic regression analysis for each of identified risk factors and few others as dependent variables with mortality as outcome and we included significant (p < 0·05) predictors for model development (Table C in S1 Appendix). Sample size estimation was done using a thumb rule of 10 events per predictor [39]. As there were 57 events in the derivation cohort, the ideal number for predictors in the model was taken to be 6 to 7. Multiple imputation analysis was planned for addressing missing data if missing data for any predictor>5%.

## Model development

The predictors finally selected for the multivariable model are enumerated in Table C in S1 Appendix. All candidate variables from the derivation cohort were entered into the multivariable logistic regression analysis. We used a backward stepwise elimination approach with the least statistically significant variable removed at each step. A total of five elimination steps simplified the model based on minimum Akaike Information Criteria (AIC) value.

## Conversion to score

In the final model, we assigned the scores proportional to their β regression coefficients of the multivariable regression equation, using standard approach [40]. The variable with minimum β value was assigned a score of 1 and the remaining variables were assigned proportional scores with rounding off to the nearest integer to generate an easily calculable score [39,40]. An arbitrary cut-off score was chosen based on the score-mortality estimate graph.

## Model performance, predictive accuracy, and external validation

Discrimination (i.e., the degree to which a model differentiates between those who died and survived) was calculated with concordance (c-index or statistic), equivalent to the area under the ROC curve. A value of 0.5 indicates no predictive ability, 0.8 is considered good, while 1 is perfect discrimination. Hosmer and Lemeshow goodness of fit statistic and Nagelkerke $r^2$ were calculated for assessing overall model performance. To assess the calibration of the model, (i.e., agreement between predicted and observed risk of mortality), calibration plots were used. Perfect calibration is implied by a 45˚ diagonal line (calibration slope = 1 and a calibration intercept = 0). Deviations above or below the line reflects underprediction and overprediction by the model. We assessed the predictive accuracy of the score in the validation cohort with discrimination and calibration as mentioned above. We did all analysis with SPSS statistical software v23. Calibration plots were constructed Stata/IC v16 (trial version). The present study is reported in compliance with standard TRIPOD guidelines for prediction models (S1 TRIPOD Checklist).

## Results

For the selection of candidate variables, 15 studies were reviewed to generate a list of 25 potential parameters. Related parameters were combined for clarity (e.g., shock and hypotension, anaemia, and haemoglobin < 10 g/dL). Ten parameters were considered infeasible for primary care settings and were excluded, among which, 3 were not deemed suitable for measurement on day 1 of bite. Two parameters reported in only a single study done on children were also not included (Table A in S1 Appendix). The derivation cohort included 248 while the

validation cohort comprised 140 participants. Baseline characteristics for both cohorts are summarized in Table 1.

In the derivation cohort, 74.1% (n = 184) and validation cohort, 79.2% (n = 119) were classified as Russell's viper envenoming by either snake identification or syndromic diagnosis (syndromes 2 & 5). Also, 19% in derivation cohort and 15% in validation cohort were classified as viper envenoming with unspecified species—syndrome 1 i.e., local swelling with prolonged WBCT20. A section of these patients is also expected to be Russell's viper envenoming.

Univariable analysis in the derivation cohort (Table B in S1 Appendix,) found a significant association of in-hospital mortality with several predictors that were consistently reported previously: systolic blood pressure <100 mm Hg, presence of signs of capillary leak syndrome (CLS), any overt bleeding manifestations at admission, severity of local swelling, bite-to-antivenom time> 6.5h, haemoglobin <10 g/dL, presence of acute kidney injury (defined as creatinine >3 mg/dL), urine output < 20 mL/hour in the first 24 hours (measured over 24 hours), urine albumin positive by dipstick and thrombocytopenia (platelet < 260 x $10^9$/L) (Table 2). These variables were entered into a multivariable model. Age and gender were also included in the model, despite being non-significant in the univariable analysis, because they were clinically relevant predictors.

Seven predictors remained in the multivariable model at step 5: overt bleeding, haemoglobin at admission <10 g/dL, bite to antivenom time> 6.5 hours, systolic blood pressure at admission < 100 mm Hg, presence of signs of capillary leak syndrome, urine output < 20 mL/hour in the first 24 hours and female gender (Tables 3 and 4). The predictors which were not significant at step 5 were also retained in the model considering optimal AIC and need to retain some clinically important predictors like bite-to-antivenom time which clinicians find valuable. Although AIC was minimum in step 6, we limited to five elimination steps in order to retain bite-to-antivenom time a clinically significant predictor variable as mentioned above based on clinician inputs and prior reports[34]. (Table 2 and Tables C and D in S1 Appendix). The regression equation and intercept (baseline mortality risk) are shown in Table 4. We

**Table 1. Clinical characteristics of Derivation and Validation cohorts.**

| Characteristics | Derivation cohort (N = 248) | Validation Cohort (N = 140) |
|---|---|---|
| **Enrolment period** | August 2011—August 2013 | September 2013—July 2015 |
| **Mean Age (SD) in years** | 40 (13–76) | 39 (12–67) |
| **Male gender (%)** | 168 (68) | 103 (74) |
| Species identification: **Snake species identified by dead snake or photograph** Russell's viper Saw scaled viper | 36 (14.5) 17 (6.85) | 8 (5.7) 3 (2.1) |
| **Syndromic diagnosis** Russell's Viper (Syndromes 2/5) Viperidae (Syndrome 1) | 148 (59.7) 47 (18.9) | 108 (77.1) 21 (14.8) |
| **Lower limb bites (%)** | 206 (83) | 119 (85) |
| **Occupational bites (Agricultural activities) (%)** | 173 (70) | 105 (75) |
| **Antivenom dose (ml)—Median (IQR)** | 310 (167–420) | 200 (100–290) |
| **Bite to antivenom (h)—Mean (SD)** | 6.0 (3–12) | 3.25 (2–6) |
| **Acute Kidney Injury (%)** | 159 (64.1) | 79 (56.4) |
| **Required renal replacement therapy (%)** | 100 (40.3) | 45 (32.1) |
| **Required surgical limb debridement (%)** | 19 (7.6) | 9 (6.4) |
| **Mortality (%)** | 57 (22.9) | 20 (14.3) |

**Table 2. Variables in the final multivariable regression model at step 5 of backward elimination with regression coefficients, adjusted odds ratio, p value, confidence intervals and points allotted in the score.**

| Parameter | β | Adjusted Odds Ratio Exp (B) | P value (95% CI) | Points allotted in VENOMS score |
|---|---|---|---|---|
| Female Gender | 0.903 | 2.467 | 0.084 (0.89–6.87) | 1 |
| CLS | 2.178 | 8.833 | < 0.0001 (3.33–23.44) | 2 |
| Bite to ASV >6.5 hours | 0.660 | 1.934 | 0.109 (0.74–5.08) | 1 |
| Bleeding | 2.848 | 17.256 | < 0.0001 (3.84–77.57) | 3 |
| Haemoglobin < 10g/dL | 0.806 | 2.238 | 0.108 (0.84–6.10) | 1 |
| Urine output < 20 ml/h | 2.173 | 8.783 | < 0.0001 (2.84–27.15) | 2 |
| SBP < 100 | 1.888 | 6.589 | < 0.0001 (2.44–17.77) | 2 |
| Constant | -7.276 | 0.001 | < 0.0001 | |

Bite to ASV: Bite to antivenom time >6.5 hours, CLS: Capillary leak syndrome, Hb: Haemoglobin < 10g/dL, SBP <100: Systolic Blood Pressure <100 mm Hg, Urine output < 20 ml/h on day 1 of admission.

**Table 3. Calculation of VENOMS score.**

| Parameter | VENOMS score points |
|---|---|
| **Gender** | |
| Female | 1 |
| Male | 0 |
| **CLS** | |
| Yes | 2 |
| No | 0 |
| **Bite to ASV time > 6.5 hours** | |
| Yes | 1 |
| No | 0 |
| **Bleeding** | |
| Yes | 3 |
| No | 0 |
| **Haemoglobin** | |
| > 10 g/dL | 1 |
| < 10 g/dL | 0 |
| **Urine output (in first 24 hours)** | |
| < 20 ml/hr | 2 |
| > 20 ml/hr | 0 |
| **Systolic BP** | |
| < 100 mm Hg | 2 |
| > 100 mm Hg | 0 |

To calculate an individual's VENOMS score, the points associated with each predictor can be added to obtain the total risk score. As an example, a female who has a presented 8 hours after snakebite with overt bleeding, Blood Pressure 120/80 mm Hg, with no signs of CLS and urine output of 10 ml/hr will have a risk score of 1 + 1 + 3 + 0 + 0 + 1 = 7 points. According to Fig 2, 7 points corresponds to a mortality risk of 22%. ASV: antivenom, BP: Blood pressure, CLS = Capillary Leak Syndrome.

**Table 4. Final model with regression equation, intercept, and regression coefficients.**

| |
|---|
| **Log(p/1-p) = -7.276 + 0.903$x_1$+ 2.178$x_2$+ 0.660$x_3$+ 2.848$x_4$+ 0.806$x_5$+ 2.173$x_6$+1.888$x_6$** |
| Log(p/1-p) = Log odds of mortality, Constant = -7.276, $X_1$: Female gender, $X_2$: Signs of increased capillary permeability, $X_{3:}$ Bite to antivenom time > 6.5 hours, $X_4$: Overt bleeding, $X_5$: Haemoglobin < 10 g/dL, $X_6$: Systolic BP < 100 mm Hg, $X_7$: Urine output < 20 ml/hour. |

assigned point values to these items and developed an integer-based estimation system (Tables 2 and 3).

## Missing data

Missing data was < 5% for the predictor variables as data collection was prospective in the derivation cohort. Of the relevant predictors, data were 99·1% complete for 2 predictors (haemoglobin, platelet count) and 97.8% for serum creatinine. Data were complete for 100% of outcome parameters in the derivation cohort. Data was 100% complete for predictors and outcomes in the validation cohort as it was a randomized trial. As missing data was <5% we did not perform multiple imputation analysis.

## Internal validation, discrimination, and calibration

Mortality risk plotted against each point of the score showed a sigmoid curve with steep increase in mortality when score was greater than 6 (Fig 2A). Hence, we decided to take a score of 6 as a cut-off for poor prognosis. Model discrimination using a ROC showed Area Under Curve (AUC/c-index) of 0.948 (95% CI 0.92–0.98) suggesting excellent discrimination. A cut-off of 6 as discussed above had a sensitivity of 90% and specificity of 83% for predicting

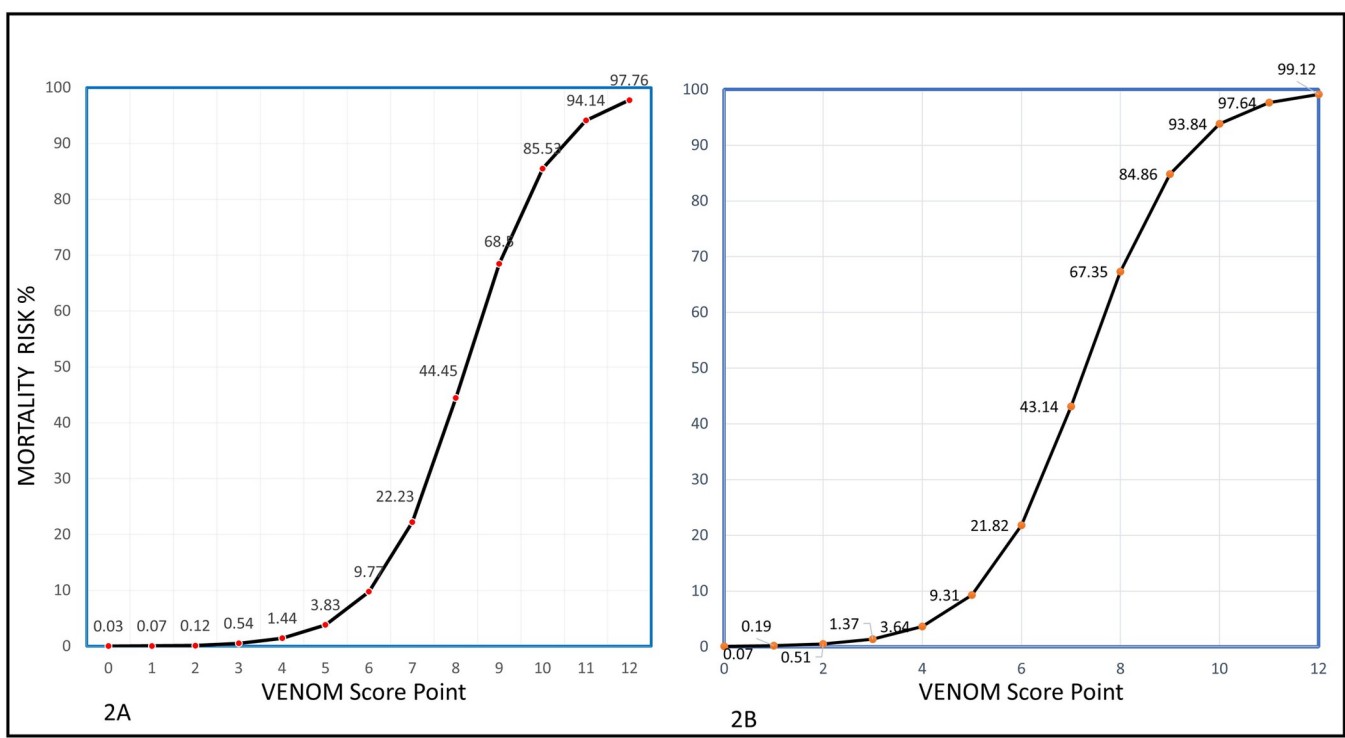

**Fig 2. A:** Mortality risk plotted against each point of the score for the derivation cohort (n = 248) showing a sigmoid curve with steep increase in mortality at score was greater than 6. **B:** Mortality prediction estimates for validation cohort (n = 140).

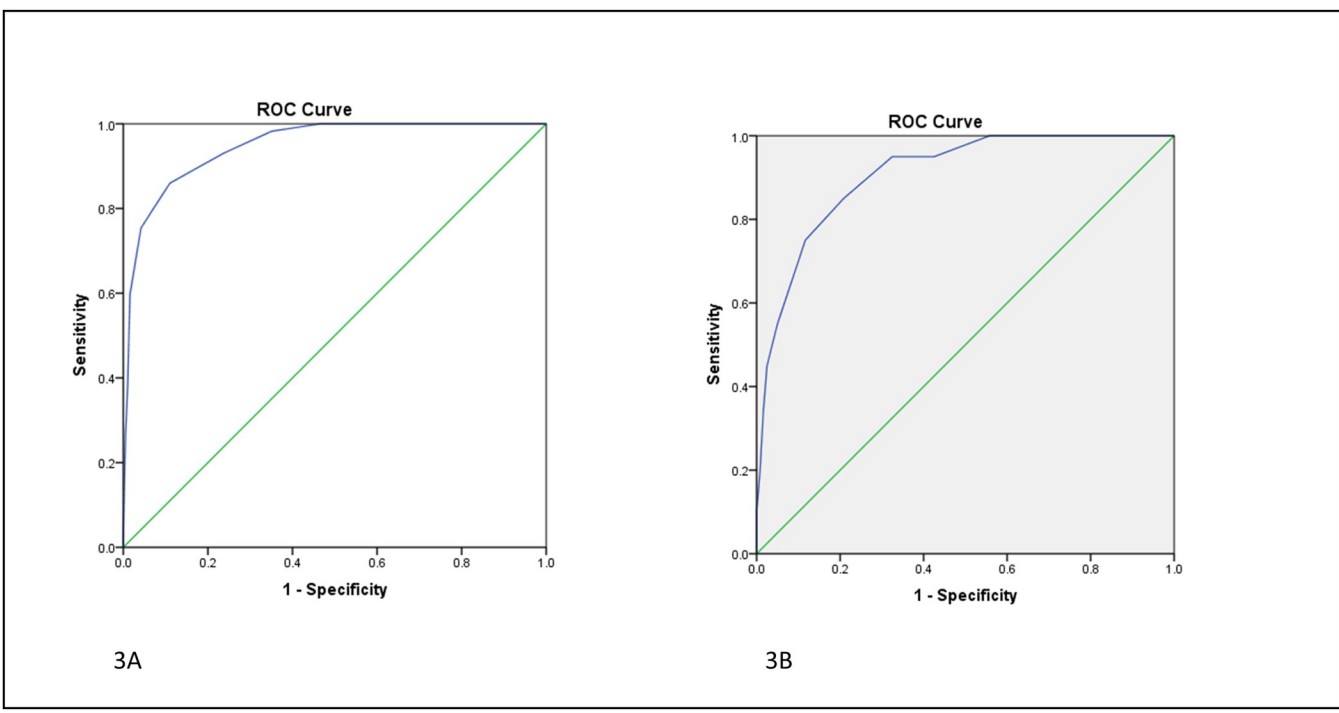

**Fig 3. A:** Model discrimination in derivation cohort using a receiver operator characteristic curve (ROC) showing area Under Curve (AUC/c-index) of 0.948 (95% CI 0.920–0.976). A cut-off of 6 had a sensitivity of 90% and specificity of 83% for predicting mortality. **B:** Model performance in validation cohort using a ROC showing AUC/c-index of 0·90 (95% CI 0·85–0·97).

mortality (Fig 3A). Hosmer-Lemeshow goodness of fit showed a chi-squared statistic of 1.52 (p = 0.99, df = 8) suggesting a good model fit. Nagelkerke $r^2$ at step 5 was 0.69 again suggesting that the model explained 70% of the variability in the outcome parameter and a good overall performance (Table D in S1 Appendix). Internal calibration showed a slope of 1, intercept of 0 and an AUC of 0.95 suggesting excellent calibration in the derivation dataset (Fig A in S1 Appendix).

## External validation

The score was a significant predictor of mortality in the validation cohort (Odds ratio [OR] 1·8 per unit increase in score, 95% CI; p < 0·0001). Model performance in the validation cohort showed a c-statistic of 0·90 (95% CI 0·85–0·97) (Fig 2B). The model predicted a mean probability of mortality as 11% (95% CI 8–15%) in the validation cohort. Thus the 95% CI included the actually observed mortality of 14.3% indicating that calibration at large was satisfactory. Calibration plots of predicted and observed mortality showed a slope of 0.7, intercept of 0.4 and a c-index (AUC) of 0.92 suggesting overall overfitting of the model within the validation cohort with overprediction at low-risk patients and underprediction of mortality in high-risk patients (Fig 4). Prediction estimates in validation cohort are shown in Fig 2B. In the validation cohort, the lowest score of 0 predicted a mortality risk of 0.06% while a score of 12 predicted a mortality of 99.1%. Sensitivity, specificity positive and negative predictive values (PPV and NPV) at each point in the score was calculated for the validation cohort and is presented in Table 5. At the selected cut-off of 6 the sensitivity was 75%, specificity 88.3%, PPV 52% and NPV 96% in the validation cohort.

**Fig 2B:** Mortality prediction estimates for validation cohort (n = 140).

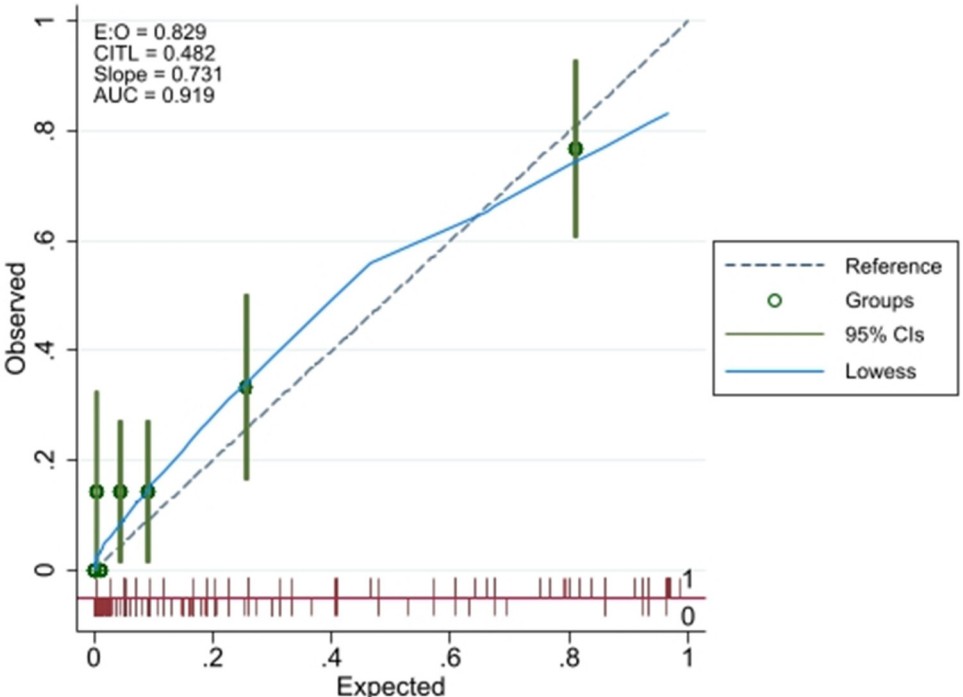

**Fig 4. Predicted versus observed mortality risk in the validation cohort.** Calibration plots showing a slope of 0.7, intercept (CITL) of 0.4 and a c-index (AUC) of 0.92. E:O: ratio of expected to observed mortality. Graph created using pmcalplot in STATA, Stata/IC 16 for Windows.

**Fig 3B:** Model performance in validation cohort using a ROC showing AUC/c-index of 0·90 (95% CI 0·85–0·97).

## Discussion

Snakebite envenoming usually affects those living in rural areas and in poverty [1,2,6]. Ending this neglect requires a refocus of research efforts into various aspects of snakebite envenoming including prognostic models to help classify patients according to severity and help plan appropriate management.

In this study, we have developed a practical prognostic instrument to predict the risk of in-hospital mortality after viper envenoming. The VENOMS score calculated on the day of admission was successfully externally validated and showed good discrimination and reasonable calibration in the same settings. The model incorporates seven items: overt bleeding manifestations, presence of signs of capillary leak syndrome, systolic blood pressure <100 mm Hg, urine output < 20 mL/h over first 24 hours (assessed over 24 hours), haemoglobin <10 g/dL, female gender, and bite to ASV time >6.5 hours. We prudently selected a list of candidate predictors and categorized them in the derivation cohort. Such a process involves making compromises, such as the exclusion of parameters that are not routinely assessed in a primary care clinical setting or that are not supported by sufficient validation data. The derivation cohort was adequately powered to show a good discrimination of the model. This is indicated by the 95% CIs of concordance statistics, which exceeded 0.8 in this cohort. Development and validation of the score followed established TRIPOD recommendations [41].

Prognostic scores support and improve the clinical decision making process and impact care by empowering clinicians to make evidence based decisions thereby improving patient

**Table 5. Accuracy of VENOMS score in predicting mortality in the validation cohort of patients with viper envenomation (n = 140).**

| VENOMS Score cutoff | Total number of patients corresponding to the cutoff | Among total patients, number of patients who died | Accuracy of score cut-off in predicting mortality among viper envenomed patients | | | |
|---|---|---|---|---|---|---|
| | | | Sensitivity (95% CI) | Specificity (95% CI) | Positive predictive value (95% CI) | Negative predictive value (95% CI) |
| ≥ 0 | 140 | 20 | 100 (83.2–100) | 0 (0–3.0) | 14.3 (14.3–14.3) | - |
| ≥ 1 | 110 | 20 | 100 (83.2–100) | 25.0 (17.6–33.7) | 18.2 (16.7–19.8) | 100 (100–100) |
| ≥ 2 | 87 | 20 | 100 (83.2–100) | 44.2 (35.1–53.5) | 23.0 (20.3–25.9) | 100 (100–100) |
| ≥ 3 | 70 | 19 | 95.0 (75.1–99.9) | 57.5 (48.2–66.5) | 27.1 (22.8–32.0) | 98.6 (91.0–99.8) |
| ≥ 4 | 58 | 19 | 95.0 (75.1–99.9) | 67.5 (58.4–75.6) | 32.8 (27.0–39.1) | 98.8 (92.– 99.8) |
| ≥ 5 | 42 | 17 | 85.0 (62.1–96.8) | 79.2 (70.8–86.0) | 40.5 (31.4–50.2) | 96.9 (91.7–98.9) |
| ≥ 6 | 29 | 15 | 75.0 (50.9–91.3) | 88.3 (81.2–93.5) | 51.7 (38.1–65.1) | 95.5 (90.8–97.9) |
| ≥ 7 | 17 | 11 | 55.0 (31.5–76.9) | 95.0 (89.4–98.1) | 64.7 (43.3–84.1) | 92.7 (88.6–95.4) |
| ≥ 8 | 12 | 9 | 45.0 (23.1–68.5) | 97.5 (92.9–99.5) | 75.0 (47.0–91.0) | 91.4 (87.8–94.1) |
| ≥ 9 | 9 | 7 | 35.0 (15.4–59.2) | 98.3 (94.1–99.8) | 77.8 (43.9–94.0) | 90.1 (86.8–92.6) |
| ≥ 10 | 5 | 4 | 20.0 (5.7–43.7) | 99.2 (95.4–99.98) | 80.0 (32.0–97.1) | 88.2 (85.7–90.3) |
| ≥ 11 | 2 | 2 | 10.0 (1.2–31.7) | 100 (97.0–100) | 100 (100–100) | 87.0 (85.2–88.5) |
| 12 | 0 | 0 | 0 (0–16.8) | 100 (97.0–100) | - | 85.7 (85.7–85.7) |

outcomes[39]. Classical examples include Wells score for predicting pulmonary embolism and CURB 65 or pneumonia severity index for community acquired pneumonia. Both these scores have gained widespread applicability and have resulted in impacting diagnosis and management of these conditions including reduction in mortality of admitted patients in emergency departments [42,43]. Limited clinical prediction scores are available for neglected tropical diseases [44] A commonly reported score for snakebites is the Snakebite Severity Score (SSS) which ranges from 0 to 23 and assesses respiratory, cardiovascular, hematologic, gastrointestinal, central nervous system and local wound to assign scores for each [45]. The SSS was originally evolved for evaluating dry bites and deciding if patient requires antivenom or not. SSS has been shown to limit antivenom and other resource utilization [46,47]. It has been used as a prognostic score for haemotoxic bites in Indian settings, but a formal validation is unavailable [48]. The SSS has several limitations: it combines both neurotoxic and hemotoxic manifestations, includes several laboratory results including PT, aPTT, serum fibrinogen which are usually not available at primary care settings and common elapid neurological signs like ptosis do not figure in the score [49]. Apart from the SSS, studies from Korea have used the International Society of Thrombosis and Haemostasis scoring system for disseminated intravascular coagulation to classify viper bite patients with VICC though prognostic implications were unclear [50,51]. Another prognostic score is the Zululand Severity Score developed in South Africa for determining whether the patient requires antivenom or surgical intervention [52]. A species-specific severity grading for Indian snakes was evolved by Kumar V et al and was reported in subsequent hospital based studies [53,54]. However, the score is complex, the basis

for severity grading are unclear and its prognostic implications were not validated. Patient-Specific Functional Scale (PSFS), is a patient-reported outcome that is validated for assessing limb recovery from snakebite envenoming [55]. In summary, there exists a need for a simple bedside prognostic instrument which can help triage and appropriately manage viper envenoming patients.

The VENOMS score has several potential practical applications despite being currently validated in a single centre: it can be applied readily at the bedside by clinicians without any device to stratify viper envenoming patients. We expect that the score can help tailor care according to risk-class by triaging low and high mortality risk (score >6) patients who may require early intensive care. We hypothesize that the score might aid decision making for early transfers while reducing unnecessary referrals in primary care settings. We also suspect that the score has a potential to reduce antivenom overuse in the form of additional doses in patients with low VENOMS score (e.g., a cut-off < 4 have mortality of 1.5%) similar to the SSS [46]. However, further clinical studies are warranted to confirm these suggestions. Cost-effectiveness and acceptability of VENOMS score also need further research. Likewise, the study opens several interesting questions which need further exploration in clinical context such as what are appropriate measures to reduce mortality, in high-risk individuals (Score >6) and what is performance of the score as a guide to supportive care?

Our study has several important limitations. A syndromic approach to identifying the offending snake may have resulted in errors. The scoring system has only been validated in the same centre as the derivation cohort, where the common species is Russell's viper (at least 74% patients in derivation cohort and 79% in validation cohort fitted into confirmed or syndromic diagnosis of Russell's viper). The score requires independent external validation in other settings before widespread applicability. The performance of this score in settings where saw-scaled viper envenoming forms bulk of cases will need appropriate modification of the score. The scope of the score is limited to in-hospital mortality.

Clinical manifestations vary greatly across India and South Asia, and our sample is from a single site. Geographical intraspecific variations in Russell's viper envenoming has been known to cause varied clinical manifestations [12]. For example, capillary leak syndrome due to *Daboia russelii* envenoming has been frequently reported from Southern India, Sri Lanka, and Myanmar while there are only few reports of this phenomenon in from other areas in the subcontinent [19]. Likewise, pre-synaptic neurotoxic features in Russell's viper envenoming appear to have limited geographical distribution [23]. Therefore, apart from the spectrum effect in clinical prediction scores, the score requires further widespread geographical as well as domain validation specifically in primary care settings.

All predictors were converted to categorical variables for ease of use, this might have led to some loss of information. There were some differences in baseline characteristics of both the cohorts even though they were from the same centre. This difference could be attributed to differences in study design (prospective cohort *vs* randomized clinical trial) and inclusion and exclusion criteria. Specifically, the validation cohort excluded patients who had received > 20 vials antivenom prior to admissions. It is possible that some severely envenomed patients (who are likely to receive higher doses of antivenom upfront at primary care settings) were missed in the derivation cohort. Also, even though both cohorts received antivenom from the same manufacturer (Table 1), multiple batch numbers were used according to institutional supply which might have resulted in varying action due to batch to batch variation [56,57]. It is pertinent to note that the median antivenom dose used by the derivation cohort is 30 vials which is the recommended upper limit for Russell's viper envenoming suggesting that many patients received more antivenom than recommended but did not respond as expected. Also,

the results are only applicable to adults >12 years of age as we did not include children who may have different clinical predictors as suggested by previous studies.

Selection bias needs to be considered because both cohorts pre-selected people with severe envenoming and the population was a tertiary care referral centre [39]. Both cohorts used clinical syndromic approach to snake identification based on the current WHO guidelines while serum-based assays could have ascertained species-based diagnosis of viper envenoming. However, this approach mimics a real-life situation, including rural primary care scenarios, possibly making the model applicable in these practice settings. There was deviation from the perfect slope in validation calibration plot (Fig 4). These deviations were limited in scope and within the estimated 95% CI. Also, smoothing techniques used to estimate the observed probabilities of the outcome in relation to the predicted probabilities, i.e. the loess algorithm may have affected the graphical impression, considering that the derivation cohort is a smaller dataset [58].

In conclusion despite limitations, the VENOMS score appears to be an easy-to-use point of care clinical prediction score for mortality prediction for Russell's viper envenoming in Southern India with potential widespread applications in various settings.

## Supporting information

**S1 TRIPOD Checklist. TRIPOD checklist for prediction model development.**
(DOCX)

**S1 Appendix Text. Search Strategy, Potential variables considered and references. Table A:** Publications screened for variable selection for model development **Table B:** (Supplementary Appendix 1): Area under the curve (AUC) for Receiver-operating curves (ROC) constructed for continuous predictor variables with mortality (or survival) as the state variable with confidence interval (CI), cut off chosen and sensitivity and specificity at chosen cut-off. **Table C:** Odds ratio with 95% CI for univariable Binary Logistic Regression (unadjusted) and subsequent multivariable logistic regression with backward elimination strategy (adjusted) to predict mortality as outcome. **Table D:** Multivariable logistic regression model with backward elimination at step 5 and step 7 **Table E:** Model summary showing -2 log likelihood, Cox and Snell's R square, Nagelkerke R Square and Akaike Information criteria (AIC) shown for each step of backward elimination. **Fig A:** Perfect Internal calibration in derivation cohort (slope of 1, intercept of 0 and an AUC of 0.95). Graph created using pmcalplot in STATA, Stata/IC 16 for Windows.
(DOCX)

**S1 Data. Deidentified patient data for derivation cohort.**
(PDF)

**S2 Data. Deidentified patient data for validation cohort.**
(PDF)

## Acknowledgments

We gratefully acknowledge Dr L. Jeyaseelan, Professor and Head, Department of Biostatistics, Christian Medical College, Vellore for his guidance in multivariable logistic regression and score assignment.

## Author Contributions

**Conceptualization:** Maya Gopalakrishnan, Suman Saurabh.

**Data curation:** Maya Gopalakrishnan, Pramod Sagar, Chanaveerappa Bammigatti, Tarun Kumar Dutta.

**Formal analysis:** Maya Gopalakrishnan, Suman Saurabh, Chanaveerappa Bammigatti.

**Investigation:** Pramod Sagar, Chanaveerappa Bammigatti.

**Methodology:** Maya Gopalakrishnan, Suman Saurabh, Pramod Sagar, Chanaveerappa Bammigatti, Tarun Kumar Dutta.

**Project administration:** Maya Gopalakrishnan, Suman Saurabh, Chanaveerappa Bammigatti, Tarun Kumar Dutta.

**Software:** Maya Gopalakrishnan, Suman Saurabh.

**Supervision:** Maya Gopalakrishnan, Chanaveerappa Bammigatti, Tarun Kumar Dutta.

**Validation:** Maya Gopalakrishnan, Suman Saurabh.

**Writing – original draft:** Maya Gopalakrishnan, Suman Saurabh.

**Writing – review & editing:** Maya Gopalakrishnan, Suman Saurabh, Pramod Sagar, Chanaveerappa Bammigatti, Tarun Kumar Dutta.

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
