## [Decision Letter · Decision Letter 0]

27 Jul 2021

Dear Dr. Maya Gopalakrishnan,

Thank you very much for submitting your manuscript "A simple mortality risk prediction score for viper envenoming in India (VENOMS): A model development and validation study" for consideration at PLOS Neglected Tropical Diseases. As with all papers reviewed by the journal, your manuscript was reviewed by members of the editorial board and by several independent reviewers. In light of the reviews and the editor's comments (below this email), we would like to invite the resubmission of a significantly-revised version that takes into account the reviewers' comments. 

We cannot make any decision about publication until we have seen the revised manuscript and your response to the reviewers' comments. Your revised manuscript is also likely to be sent to reviewers for further evaluation.

Sincerely,

Nicholas R Casewell

Associate Editor

Janaka de Silva

Deputy Editor

Editor's comments:

1. A syndromic approach to identifying the offending snake may well have resulted in errors. 

2. Clinical manifestations vary greatly across India, and this sample of bite victims is from a single site. 

3. The scoring system has only been validated in a single centre and that this was the same centre as the derivation cohort. The score requires independent external validation in other settings. 

The authors should acknowledge these major limitations in the methodology, and the discussion and conclusions should take these limitations into account (be less dogmatic).

Reviewer's Responses to Questions

**Key Review Criteria Required for Acceptance?**

**Methods**

-Are the objectives of the study clearly articulated with a clear testable hypothesis stated?

-Is the study design appropriate to address the stated objectives?

-Is the population clearly described and appropriate for the hypothesis being tested?

-Is the sample size sufficient to ensure adequate power to address the hypothesis being tested?

-Were correct statistical analysis used to support conclusions?

-Are there concerns about ethical or regulatory requirements being met?

Reviewer #1: Abstract line 26 and and intro line 124 - is there conclusive evidence to support that mortality is higher amongst viperid envenomings? I would have thought neurotoxic elapdis would have a higher case fatality rate?

Author summary line 55-57 - it is too early to suggest the score can be used to inform clinical practice. It has only been validated in a single centre, and that was the same centre as the derivation cohort. Suggest reword to, 'may become' or 'shows promise'.

Methods line 155 - specify if patients were enrolled consecutively for the derivation cohort?

Methods Table 1 - Although this is based on previously published data, suggest moving this table to the results section. 

Methods table 2 - suggest removing this table. All of these covariates with corresponding ORs and 95% CIs should be listed in Table 3 of the results.

Choice of covariates - it seems unusual not to include biting species as a covariate as this seems like an important predictor of outcome. Was this not used due to missing data - in which case specify this in the paper? If there is too much missing data, in those cases where the species was known it would be helpful to present the mortality rate by species. This will show if, for example, those with known Russell's envenoming have a higher mortality than those with known Echis envenoming.

Reviewer #2: -Are the objectives of the study clearly articulated with a clear testable hypothesis stated? Yes

-Is the study design appropriate to address the stated objectives? Yes - however methodological concern as outlined below.

-Is the population clearly described and appropriate for the hypothesis being tested? Some concern - See below

-Is the sample size sufficient to ensure adequate power to address the hypothesis being tested? - Yes

-Were correct statistical analysis used to support conclusions? - Yes 

-Are there concerns about ethical or regulatory requirements being met? - No concerns

**Results**

-Does the analysis presented match the analysis plan?

-Are the results clearly and completely presented?

-Are the figures (Tables, Images) of sufficient quality for clarity?

Reviewer #1: Results Table 3 - Include all covariates that were entered into the logistic regression model. Include unadjusted (univariate) and adjusted (multivariate) ORs, 95% CIs and p values. 

Results Table 3 - there is an error in the 'Hb 10' covariate. P value is <0.0001 yet 95% CI 0.84-6.10. If this covariate is a non-significant predictor I would suggest removing it from the scoring system.

Results Table 4 - I would disagree with including 'gender' and 'bite to ASV' as predictors in the model. The confidence intervals cross 1 and therefore it is not clear if they are associated with a higher or lower risk of death. One would expect the model to be slightly more accurate without them?

Results line 306 - was multiple imputation necessary when 97.8% of data was available. Was missing data 'missing at random?' Unless data was not missing at random and this influenced the model, I would suggest not doing multiple imputation. I'm not a statistician so happy to be corrected if there is something I am missing. 

Results line 331 - request that the sensitivity, specificity, PPV and NPV is calculated for every score cut-off in the validation cohort and that this be included in an additional table. See Abouyannis et al 2011 Table 3 for an example (doi:10.1097/QAD.0b013e328349a414). PPV and NPV are particularly meaningful when applied to clinical decision making.

Reviewer #2: -Does the analysis presented match the analysis plan? Yes

-Are the results clearly and completely presented? Yes

-Are the figures (Tables, Images) of sufficient quality for clarity? In the most part. Figure 1 could be improved.

**Conclusions**

-Are the conclusions supported by the data presented?

-Are the limitations of analysis clearly described?

-Do the authors discuss how these data can be helpful to advance our understanding of the topic under study?

-Is public health relevance addressed?

Reviewer #1: More emphasis should be placed on the fact that this scoring system has only been validated in a single centre and that this was the same centre as the derivation cohort. Highlight that the tool requires independent external validation in other settings with a high burden of Russell's viper envenoming.

Reviewer #2: -Are the conclusions supported by the data presented? Some modification required

-Are the limitations of analysis clearly described? - Yes, these could be grouped together better, see comments below.

-Do the authors discuss how these data can be helpful to advance our understanding of the topic under study? Yes, but can be improved.

-Is public health relevance addressed? Yes

**Editorial and Data Presentation Modifications?**

Reviewer #1: Appendix S3 is difficult to interpret as it shows the raw output from the logistic regression.

Reviewer #2: Running title – This is misleading – needs to specify Indian Vipers.

Line 26 - Reported mortality is higher in viper envenoming compared to snakes 

– please correct.

Author summary

Line 57-58 It can also aid better decision making with respect to antivenom administration and other supportive care. The findings of this study do not support this statement, VENOMS tool was not validated as to whether patients should be administered antivenom/repeated antivenom

Image quality of Figure 1 is poor.

**Summary and General Comments**

Reviewer #1: Overall I think the topic area and the findings of this paper are highly important and certainly worthy of acceptance once the above changes have been made. Particularly in India, where mortality rates from envenoming are high, there is a great need for validated scoring systems that clincians can use to help inform their decisions. The major limitation is that this is single centre, and validated in the same centre, so generalisibility is not confirmed.

Reviewer #2: The authors should be congratulated for devising a severity scoring system using a systematic approach to development and validation for Viper envenoming in India. Some suggestions and concerns are outlined below. 

Methodological concerns

Syndromic approach to identifying snake may have resulted in errors in inclusion/exclusion 

Sampling was from one site only – we know that clinical manifestations vary greatly across India and Indian subcontinent. 

Validation cohort selected limited patients with >200ml antivenom prior to presentation. This is likely to bias the patient selection to less severely envenomed patients (demonstrated in Table 1 with reduced complication rates between groups 1 and 2 and a reduced mortality rate for the validation group).

Smaller concerns

Authors need to define CLS and the time at which data points were collected – Was this at admission or at 24 hours? (Bearing in mind the median time to development of signs is 48 hours).

Results

Did the authors assess heart rate as a potential variable? I note this was not included in the list of 25 potential variables listed in the supplementary material. 

After selection of the variables for inclusion in the VENOMS score – I would like to see a comparison of mortality risk curves for data collected at different time points. E.g on admission, admission +12 hours, Admission +24 hours, Admission + 36 hours, Admission + 48 hours. If the time points mortality risk scores are not statistically significant for each time point, then it supports the statement that the score can be calculated at any point between presentation and 48 hours after.

Discussion 

I believe the paragraph from Lines 438-449 should be communicated early in the discussion. 

Please expand on how use of the VENOMS severity score may improve snakebite management? Are there examples in the literature? How have clinical severity scoring for other conditions improved care (such as CURB65)?

As mentioned in comments, I believe the subjects outlined between Lines 396-412 are limitations of this study methodology and that practical prospective assessment of this assessment tool is needed from multiple study sites (external validation) to corroborate the study findings.

Data sharing

All individual patient data sets should be made readily available to meet PLoS NTD data sharing statement. This is not presently provided.

PLOS authors have the option to publish the peer review history of their article (what does this mean?). If published, this will include your full peer review and any attached files.

Reviewer #1: Yes: Michael Abouyannis

Reviewer #2: No
---

## [Decision Letter · Decision Letter 1]

14 Dec 2021

Dear Dr Gopalakrishnan,

Thank you very much for submitting your manuscript "A simple mortality risk prediction score for viper envenoming in India (VENOMS): A model development and validation study" for consideration at PLOS Neglected Tropical Diseases. As with all papers reviewed by the journal, your manuscript was reviewed by members of the editorial board and by several independent reviewers. The reviewers appreciated the attention to an important topic. Based on the reviews, we are likely to accept this manuscript for publication, providing that you modify the manuscript according to the review recommendations. 

The reviewers both acknowledged the considerable effort undertaken in the revision submitted, and the resulting impact that this has had on the manuscript. Please attend to the minor revisions suggested by reviewer one (and the reference corrections indicated by reviewer two). We thank the authors for engaging productively in the review process.

Sincerely,

Nicholas R Casewell

Associate Editor

Janaka de Silva

Deputy Editor

The reviewers both acknowledged the considerable effort undertaken in the revision submitted, and the resulting impact that this has had on the manuscript. Following the completion of the minor revisions suggested by reviewer one (and the reference corrections indicated by reviewer two) this manuscript will be accepted for publication. I thank the authors for engaging to productively in the review process.

Reviewer's Responses to Questions

**Key Review Criteria Required for Acceptance?**

**Methods**

-Are the objectives of the study clearly articulated with a clear testable hypothesis stated?

-Is the study design appropriate to address the stated objectives?

-Is the population clearly described and appropriate for the hypothesis being tested?

-Is the sample size sufficient to ensure adequate power to address the hypothesis being tested?

-Were correct statistical analysis used to support conclusions?

-Are there concerns about ethical or regulatory requirements being met?

Reviewer #1: Yes

Reviewer #2: (No Response)

**Results**

-Does the analysis presented match the analysis plan?

-Are the results clearly and completely presented?

-Are the figures (Tables, Images) of sufficient quality for clarity?

Reviewer #1: Table 5 - I suggest making it clear that each row refers to a score cut-off. Perhaps rename 1st column as “Venom score cut-off” and add ‘≥’ in front of the 0-12. It is also a bit confusing that for sens/spec/ppv/npv the accuracy using the cut-off is used, whereas for number of patient and number of deaths the number with that specific score is used. Perhaps change column 2 and 3 to the number of patients with that cut-off (i.e., the first row (score cut-off ≥0) would be 140 patients and 20 deaths. Second row 110 patients and 20 deaths). It may be possible to do this a different way - it just needs to be more clear and consistent for the reader.

Table 5 - I think there may be errors in the values in this table. For score cut-off of ≥2 the sensitivity is 97.5%. But no deaths are missed at this cut-off so sensitivity should be 100%? With a score cut-off of ≥3 the sensitivity is 95% (seems correct as 1 of 20 deaths missed). Score cut-off of ≥4, 90% sensitivity but this should be 95% as still only one death missed? I haven’t checked spec/PPV and NPV but I would suggest double checking all of the values in Table 5.

Reviewer #2: (No Response)

**Conclusions**

-Are the conclusions supported by the data presented?

-Are the limitations of analysis clearly described?

-Do the authors discuss how these data can be helpful to advance our understanding of the topic under study?

-Is public health relevance addressed?

Reviewer #1: (No Response)

Reviewer #2: (No Response)

**Editorial and Data Presentation Modifications?**

Reviewer #1: Typo with full stop mid sentence line 406

Reviewer #2: (No Response)

**Summary and General Comments**

Reviewer #1: An excellent study that has adopted a data driven apporach to developing a risk score for predicting mortality outcome of people with viper envenoming in India. With minor changes, as highlighted above, I would suggest this study be accepted.

Reviewer #2: The authors should be congratulated on the considerable improvement to the manuscript, particularly in reference to its potential applicability to clinical use. I look forward to seeing it validated in a multi-centre prospective study!

Please double check all citations correspond to the relevant reference (for example citation 38 does not correspond to reference 38 - (a letter in reference to another article)) and ensure that the list of references are referenced in accordance with journal specifications.

PLOS authors have the option to publish the peer review history of their article (what does this mean?). If published, this will include your full peer review and any attached files.

Reviewer #1: Yes: Michael Abouyannis

Reviewer #2: No

Figure Files:

Data Requirements:

Reproducibility:

References

---

## [Editor Report · Decision Letter 2]

20 Jan 2022

Dear Dr Gopalakrishnan,

We are pleased to inform you that your manuscript 'A simple mortality risk prediction score for viper envenoming in India (VENOMS): A model development and validation study' has been provisionally accepted for publication in PLOS Neglected Tropical Diseases.

Best regards,

Nicholas R Casewell

Associate Editor

Janaka de Silva

Deputy Editor

Following the final minor changes made in response to the reviewers, the revised manuscript has now been accepted for publication.

---

## [Editor Report · Acceptance letter]

15 Feb 2022

Dear Dr Gopalakrishnan,

We are delighted to inform you that your manuscript, "A simple mortality risk prediction score for viper envenoming in India (VENOMS): A model development and validation study," has been formally accepted for publication in PLOS Neglected Tropical Diseases.

Best regards,

Shaden Kamhawi

co-Editor-in-Chief

Paul Brindley

co-Editor-in-Chief
